# Precipitable Water Vapor Retrieval Based on DPC Onboard GaoFen-5 (02) Satellite

**Chao Wang** [1], **Zheng Shi** [2,3,*], **Yanqing Xie** [4], **Donggen Luo** [5], **Zhengqiang Li** [2,3], **Decheng Wang** [1] **and Xiangning Chen** [1]

1. Space Information Academy, Space Engineering University, Beijing 101416, China
2. State Environment Protection Key Laboratory of Satellite Remote Sensing, Aerospace Information Research Institute, Chinese Academy of Sciences, Beijing 100094, China
3. University of Chinese Academy of Sciences, Beijing 100049, China
4. Shanghai Institute of Satellite Engineering, Shanghai 201109, China
5. Anhui Institute of Optics and Fine Mechanics, Hefei Institutes of Physical Science, Chinese Academy of Sciences, Hefei 230031, China
* Correspondence: shizheng@radi.ac.cn

**Abstract:** GaoFen-5 (02) (GF5-02) is a new Chinese operational satellite that was launched on 7 September 2021. The Directional Polarimetric Camera (DPC) is one of the main payloads and is mainly used for the remote sensing monitoring of atmospheric components such as aerosols and water vapor. At present, the DPC is in the stage of on-orbit testing, and no public DPC precipitable water vapor (PWV) data are available. In this study, a PWV retrieval algorithm based on the spectral characteristics of DPC data is developed. The algorithm consists of three parts: (1) the construction of the lookup table, (2) the calculation of water vapor absorption transmittance (WVAT) in the band at 910 nm, and (3) DPC PWV retrieval. The global PWV results derived from DPC data are spatially continuous, which can illustrate the global distribution of water vapor content well. The validation based on the Aerosol Robotic Network (AERONET) PWV data shows that the DPC PWV data have accuracy similar to that of Moderate-resolution Imaging Spectroradiometer (MODIS) PWV data, with coefficient correlation of determination ($R^2$), mean absolute error (MAE), and relative error (RE) of 0.32, 0.30, and 0.93 using the DPC and 0.23, 0.36, and 0.96 using the MODIS, respectively. The results show that our proposed DPC PWV retrieval algorithm is feasible and has high accuracy. By analyzing the errors, we found that the calibration coefficients of the DPC in the 865 nm and 910 nm bands need to be updated.

**Keywords:** GF5-02 satellite; Directional Polarimetric Camera (DPC); precipitable water vapor (PWV); remote sensing retrieval





## 1. Introduction

Water vapor is one of the important sources of the greenhouse effect [1,2] and a key factor affecting precipitation and severe weather [3,4]. The main source of atmospheric water vapor is the evaporation of water and the transpiration of plants. Although water vapor makes up only a small part of the entire atmosphere, it plays important roles in the Earth's weather system and climate change. In addition, water vapor emits and absorbs radiation in a specific spectral range, which significantly affects the accuracy of quantitative remote sensing, such as aerosol property retrieval and atmospheric correction of remote sensing data [5–7]. Driven by thermodynamics, atmospheric water vapor shows distinct spatial and temporal differences in certain regions [8,9]. Precipitable water vapor (PWV) can characterize the water vapor content in the atmosphere. The PWV is the water vapor content in a vertical column per unit cross-sectional area from the ground to the top of the atmosphere. A study found that the total water vapor in the atmosphere increases by about 1% every 10 years [10]. Therefore, the satellite remote sensing of PWV has great significance

for monitoring water vapor content on a large spatial scale and for supporting studies on climate change, weather systems, and atmospheric correction of remote sensing data.

GaoFen-5 (02) (GF5-02) is a new Chinese satellite that was launched on 7 September 2021. The multi-view polarization camera Directional Polarimetric Camera (DPC) is one of the main payloads. The DPC has eight spectral observation channels in the spectral range from 0.4 μm to 0.9 μm, and it has enhanced observation information for aerosol remote sensing [11,12]. Since the DPC aerosol retrieval channels are affected by water vapor absorption, PWV data are needed to –eliminate the water vapor absorbing effect before aerosol retrieval. At present, researchers have developed many types of high-precision PWV data for different satellite sensors, such as Moderate-resolution Imaging Spectroradiometer (MODIS) PWV data and Medium Resolution Spectral Imager-2 (MERSI-2) PWV data [13–15]. However, due to the different transit times of different satellites and the significant temporal difference in the distribution of atmospheric water vapor, it is difficult to achieve accurate water vapor absorption correction of DPC data using an external PWV dataset. To solve this problem, the water vapor absorption band (910 nm) for water vapor retrieval was set in the DPC. It allows PWV results to be produced by means of DPC observation directly. In addition, as a new satellite PWV source, DPC PWV data can also provide data support for research on, e.g., climate change and weather systems.

At present, the DPC on the GF5-02 satellite is still in the on-orbit test period, and no public PWV data are available. Therefore, it is necessary and urgent to develop a DPC PWV retrieval algorithm for the production of a PWV dataset. We can roughly divide the existing water vapor retrieval algorithms into four categories according to the wavelength used: visible light algorithms, near-infrared algorithms, thermal infrared algorithms, and microwave algorithms. Visible light algorithms are developed based on the absorption characteristics of water vapor in the visible light spectrum. Such algorithms have been successfully used for water vapor retrieval with sensors such as Ozone Monitoring Instrument (OMI) and Global Ozone Monitoring Experiment-2 (GOME-2) [16,17]. Since these algorithms require observation in the visible light spectrum, they can only be implemented in cloudless areas during the daytime. Near-infrared algorithms use data from the near-infrared band, which requires the sensor to have both the water vapor absorption band and the non-absorption band in the spectral range of 850–1250 nm. Near-infrared algorithms have been applied to water vapor retrieval using satellite data, such as MODIS and MERSI-2 [13,14]. Because near-infrared algorithms perform in high-land-surface-reflectance conditions, they are only suitable for PWV retrieval in cloudless land and ocean glint areas during the daytime. Thermal infrared algorithms are developed based on thermal infrared split window channels. They are developed for and have been applied to sensors such as Advanced Very High Resolution Radiometer (AVHRR) and Spinning Enhanced Visible and Infrared Imager (SEVIRI) [18,19]. Since these algorithms only use thermal infrared data to retrieve water vapor, they can be used for both day and night water vapor detection in cloud-free areas. Microwave algorithms are suitable for sensors with microwave observation capabilities and have been successfully applied to water vapor retrieval with sensors such as Calibration Microwave Radiometer (CMR) and Microwave Radiation Imager (MWRI) [20,21]. Since microwaves have a certain penetration ability, these methods can be used for water vapor detection in both cloudless and cloudy areas, which is also a unique advantage over other water vapor retrieval algorithms.

Since the DPC has a water vapor absorption channel in the 910 nm band, only a near-infrared water vapor retrieval algorithm can be developed for water vapor retrieval with the DPC. The water vapor absorption band at 910 nm and the nearby band at 865 nm located in the atmospheric window are used to quantify the water vapor absorption transmittance (WVAT) in the DPC 910 nm channel according to the difference in the top of the atmosphere (TOA) reflectance in these two bands. PWV is then retrieved based on the calculated WVAT. We evaluated the accuracy of DPC PWV using AERONET PWV data as ground-truth data and MODIS PWV data for intercomparison. The structure of this paper is as follows: We introduce the dataset used in Section 2.1 and the retrieval strategy for DPC PWV retrieval

in Section 2.2. Then, the global distribution of DPC PWV is shown in Section 3.1. After that, we show the accuracy assessment of DPC PWV in Section 3.2, including validation against ground-based PWV and intercomparison with MODIS PWV data. In Section 4, we discuss the improvement of the existing bias of DPC PWV data. Finally, the paper is concluded in Section 5.

## 2. Materials and Methods

### 2.1. Dataset

#### 2.1.1. DPC Data

The Chinese GF5-02 satellite was launched on 7 September 2021, at the local time of descending node of 13:30. The DPC is one of seven payloads of GF5-02 and is mainly used for the retrieval of atmospheric parameters. The DPC has multi-angle observation capabilities and can achieve a maximum of 17 effective observations in different views of one target. A DPC image has a swath width of 1850 km and a spatial resolution of 1.62 km at the nadir point. The DPC has eight spectral channels with a spectral range from visible to near-infrared wavelengths. Only the channels at 490 nm, 670 nm, and 865 nm have extra polarization observation, and the other five channels at 443 nm, 565 nm, 763 nm, 765 nm, and 910 nm can only realize intensity observation. Among them, the laboratory calibration uncertainty of DPC intensity data is better than 3.94%, and the detection accuracy of polarization data is better than 0.0066. In this study, we used data from two near-infrared bands of the DPC (865 nm and 910 nm) to retrieve water vapor. The signal received in the 865 nm band is not affected by atmospheric absorption, and the signal received in the 910 nm band is strongly affected by water vapor absorption. The spectral response functions (SRFs) of these two bands and the WVAT spectrum are shown in Figure 1. The blue line in the legend indicates the WVAT when the view zenith angle and the solar zenith angle are both equal to 0° and PWV is equal to 1 cm. The yellow line is the WVAT when the zenith angle of the satellite and the sun is equal to 0° and PWV is equal to 3 cm. In this study, we used 10 days of DPC data from 20 January 2022 to 29 January 2022 provided by the China Centre for Resources Satellite Data and Application (CRESDA) during the on-orbit testing of the GF5-02 satellite.

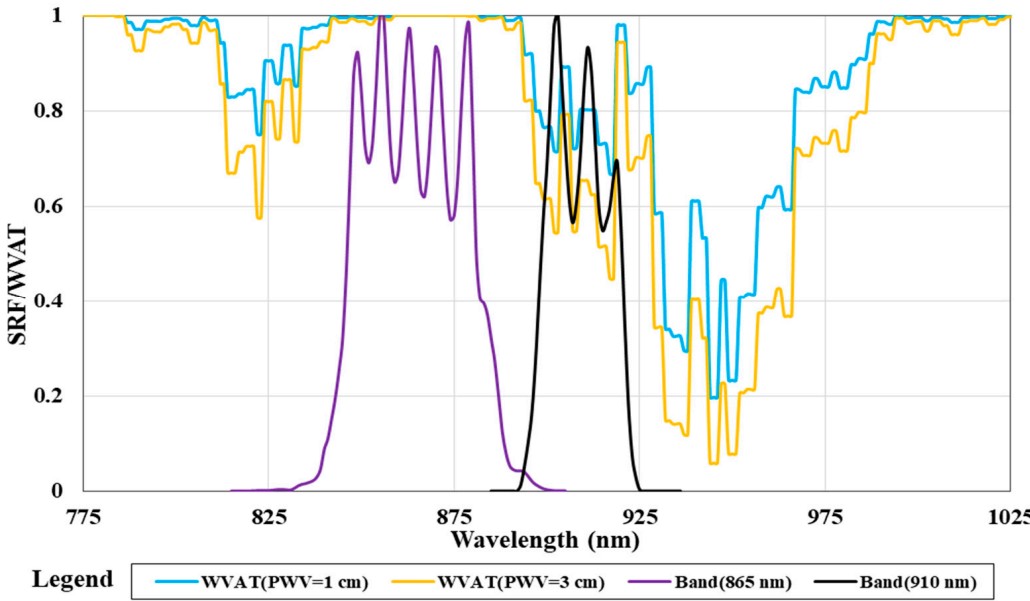

**Figure 1.** The SRFs of two near-infrared bands of DPC are illustrated by a purple line (denotes 865 nm) and a black line (denotes 910 nm), respectively. The blue line and yellow line represent the WVAT when PWV is equal to 1 and 3 cm, respectively [22].

2.1.2. MODIS PWV

The MODIS sensors on board the Terra and Aqua satellites have five near-infrared bands that can be used for water vapor retrieval: Band 2 (865 nm), Band 5 (1240 nm), Band 17 (905 nm), Band 18 (936 nm), and Band 19 (940 nm). Band 2 and Band 5 are located in the atmospheric window region and are not affected by the absorption of atmospheric gases, while Bands 17–19 are located in the spectral region of strong water vapor absorption, and the absorption effect of gases other than water vapor on these three bands is negligible. National Aeronautics and Space Administration (NASA) released official MODIS PWV data based on the above five bands. The basic principle of the official MODIS PWV retrieval algorithm is as follows: firstly, the TOA reflectance in Bands 17–19 without considering water vapor absorption is calculated using the TOA reflectance in MODIS Band 2 and Band 5 by means of linear interpolation; then, the water vapor absorption transmittance in these three water vapor absorption bands is calculated according to the TOA reflectance in Bands 17–19 and the actual TOA reflectance in Bands 17–19 when water vapor absorption is not considered. Finally, water vapor retrieval can be realized according to the obtained water vapor absorption transmittance [13]. Since we used the same PWV retrieval strategy using near-infrared channels as MODIS, and the MODIS PWV data have been extensively validated with high accuracy [23–27], we performed an intercomparison between DPC-derived PWV and MODIS official PWV data (MYD05) to comprehensively evaluate the accuracy of our developed algorithm in this research. Due to the large number of MODIS PWV data and due to the accuracies of MOD05 data developed based on Terra/MODIS data and MYD05 data developed based on Aqua/MODIS data being close [28], MYD05 data were randomly selected as the representatives of MODIS PWV data to participate in the intercomparison. The MODIS official water vapor product (MOD05/MYD05) mentioned above can be accessed at the NASA data website (https://ladsweb.modaps.eosdis.nasa.gov, accessed on 25 February 2022).

2.1.3. Ground-Based PWV Data

Aerosol Robotic Network (AERONET) is a global atmospheric aerosol observation network currently consisting of hundreds of solar photometers [29]. AERONET has been in continuous operation for more than 25 years. Although its major task is to monitor aerosol properties, it can also provide high-precision PWV data. AERONET PWV data have an uncertainty of about 10% [30,31]. Because AERONET can provide high-precision PWV data for long time series and AERONET sites are widely distributed around the world, AERONET PWV data are widely used as verification data for water vapor remote sensing retrieval results [32–34]. In this study, remote sensing PWV data were validated using level 1.5 AERONET data version 3 (data after cloud masking and quality control) [35], which can be downloaded from the AERONET homepage (https://aeronet.gsfc.nasa.gov, accessed on last accessed on 5 March 2022).

*2.2. Methods*

The PWV retrieval algorithm developed for the DPC in this study is mainly composed of three parts: (1) the construction of the DPC water vapor retrieval lookup table, (2) the calculation of WVAT in the DPC 910 nm band, and (3) DPC PWV retrieval. The flowchart of DPC PWV retrieval is shown in Figure 2.

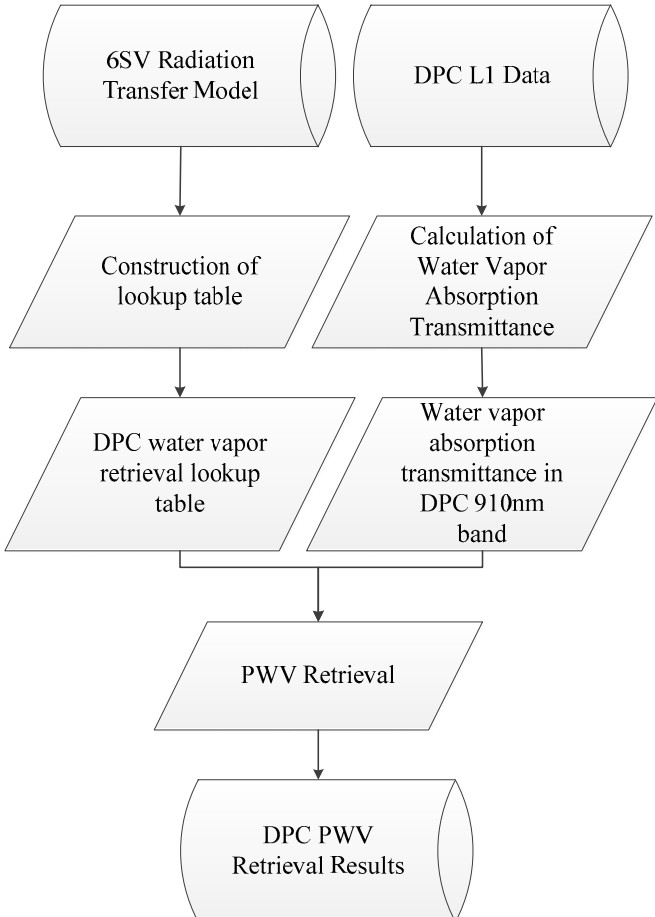

**Figure 2.** Flow chart of DPC water vapor retrieval.

### 2.2.1. Construction of DPC Water Vapor Retrieval Lookup Table

In this study, the radiative transfer model Second Simulation of a Satellite Signal in the Solar Spectrum vector (6SV) [36] was used to simulate the water vapor absorption transmittance in the DPC 910 nm band under different observed geometries and water vapor content. Then, we achieved water vapor retrieval by minimizing the simulated water vapor absorption transmittance and the actual water vapor absorption transmittance. Running 6SV in real time to calculate the water vapor absorption transmittance for PWV retrieval for each pixel is time-consuming, resulting in low retrieval efficiency. To improve the efficiency of DPC PWV retrieval, a 4-dimensional lookup table was constructed using 6SV before PWV retrieval. These four dimensions are solar zenith angle (0–72°, interval of 4°), observed zenith angle (0–72°, interval of 4°), PWV (0–8 cm, interval of 0.1 cm), and DPC 910 nm band water vapor absorption transmittance.

### 2.2.2. WVAT Calculation of DPC 910 nm

The received signal of DPC in near-infrared channels can be expressed as [10]

$$L_{sen}^{\lambda} = T_g^{\lambda} \left( L_c^{\lambda} + L_0^{\lambda} T_{\uparrow}^{\lambda} T_{\downarrow}^{\lambda} \rho_s^{\lambda} \right) \tag{1}$$

where $L_{sen}^{\lambda}$ represents the radiance received at the TOA, $T_g^{\lambda}$ represents the atmospheric absorption transmittance, $L_0^{\lambda}$ represents the solar irradiance at the top of the atmosphere, $T_{\uparrow}^{\lambda} T_{\downarrow}^{\lambda}$ represents the atmospheric total transmittance from the sun-surface–sensor path, $\rho_s^{\lambda}$ represents the surface reflectance, and $L_c^{\lambda}$ represents the atmospheric path radiance.

Dividing the two sides of Formula (1) by solar irradiance $L_0^\lambda$, the formula for calculating the TOA reflectance of the DPC near-infrared band can be obtained [14].

$$R_{TOA}^\lambda = T_g^\lambda \left( \rho_0^\lambda + T_\uparrow^\lambda T_\downarrow^\lambda \rho_s^\lambda \right) \tag{2}$$

where $R_{TOA}^\lambda$ represents the TOA reflectance and $\rho_0^\lambda$ represents the atmospheric path reflectance.

Due to the relatively close wavelengths of the DPC bands at 865 nm and 910 nm, the difference between the path reflectance and atmospheric total transmittance in these two bands can be ignored. In addition, the surface reflectance difference between the two bands is small. Based on these reasons, it can be assumed that the TOA reflectance of the DPC in the 910 nm band is equal to that in the 910 nm band without considering atmospheric absorption [22].

In addition, the influence of atmospheric gases other than water vapor on the radiation absorption of DPC Band 8 is neglectable, which means that the WVAT in DPC Band 8 is equal to the atmospheric absorption transmittance. Since the influence of the atmosphere on the absorption at DPC 865 nm can be neglected, the WVAT at 910 nm can be calculated by dividing the TOA reflectance at 910 nm by the TOA reflectance at 865 nm.

$$T_g^{910\ nm} = R_{TOA}^{910\ nm} / R_{TOA}^{865\ nm} \tag{3}$$

where $T_g^{910\ nm}$ represents the atmospheric absorption transmittance in the DPC 910 nm band, which can also be considered as the WVAT in the DPC 910 nm band, and $R_{TOA}^{865\ nm}$ and $R_{TOA}^{910\ nm}$ represent the TOA reflectance of the DPC in the 865 nm band and 910 nm band, respectively.

### 2.2.3. DPC PWV Retrieval Strategy

For any pixel to be retrieved, we can simulate 81 water vapor absorption transmittances in the 910 nm band for different water vapor contents (PWV range of 0–8 cm, interval of 0.1 cm) according to the pre-constructed lookup table and the corresponding solar zenith angle and observation zenith angle of the pixel and then calculate the water vapor absorption transmittance in the DPC 910 nm band according to Formula (3) (for convenience of description, it is called $T_1$). Then, we can realize water vapor retrieval based on the simulated and actual WVAT in the 910 nm channel by means of linear interpolation. The specific process of retrieving water vapor by means of interpolation is as follows: two adjacent water vapor absorption transmittances are found from the 81 water vapor absorption transmittances simulated above, one of which is not less than $T_1$ (called $T_2$) and the other is not greater than $T_1$ (called $T_3$), after which we can perform water vapor inversion according to Equation (4) [37].

$$V_1 = V_2 + \frac{(V_3 - V_2) \times (T_1 - T_2)}{(T_3 - T_2)} \tag{4}$$

where $V_1$ represents the PWV retrieval results, and $V_2$ and $V_3$ represent PWV corresponding to $T_2$ and $T_3$, respectively.

## 3. Results

### 3.1. Global Distribution of PWV Retrieved from DPC Data

Figure 3 shows the mean values of global water vapor from 20 January 2022 to 29 January 2022. The blank areas denote no retrieval results, which happens in cloud-contaminated regions. During the last winter season, most of the regions were dry in the Northern Hemisphere. The inland areas of North America, Europe, Asia, and northern Africa had low water vapor contents (<2 cm). Coastal and off-shore regions such as southeastern India and southwestern Africa had relatively high water vapor values, about 2 cm. The PWV contents were high (>3 cm) in Myanmar and Thailand, which have considerable rainforest coverage. Places such as Brazil, Congo, and northern Australia with

rainforest land cover had high water vapor contents. In this period, the DPC PWV data derived from our developed algorithm illustrate the climate-influenced global water vapor content distribution well.

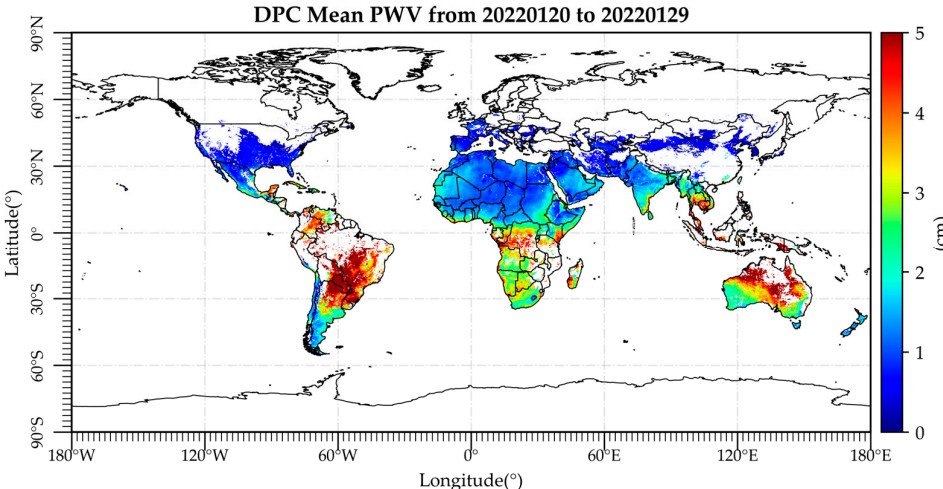

**Figure 3.** Global mean PWV values derived from DPC data for 20 January 2022 and 29 January 2022.

Since PWV has obvious zonal distribution characteristic, we calculated the zonally averaged absolute PWV values, which are depicted in Figure 4. As expected, the PWV content was higher at the mid-lower latitudes of the southern summer hemisphere. The mean PWV content was higher than 2 cm at latitudes from −40° to 10°. Two obvious peaks of PWV were found at −7° and −21°, with PWV content being higher than 4 cm. Rainfall, continuous cloud cover, and the transpiration of plants contribute to rainforest moisture in these mid-lower-latitude regions.

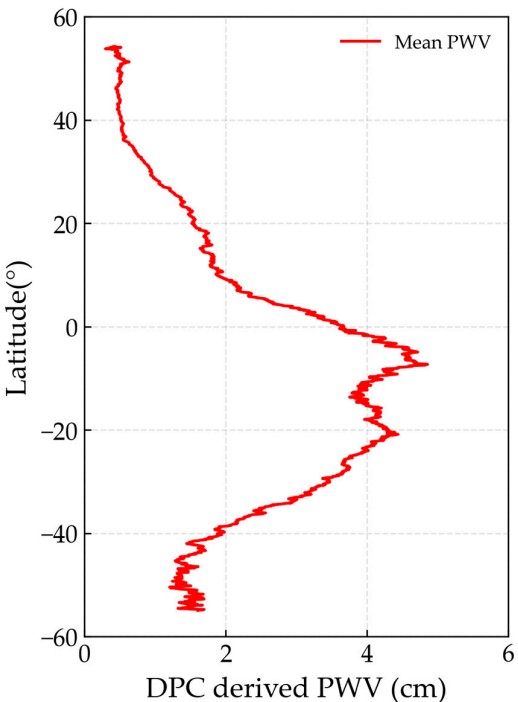

**Figure 4.** The 10-day zonally averaged PWV values observed using the DPC.

Figure 5 shows the variation in the statistical results of global mean PWV derived using the DPC. The highest global mean PWV value was 2.032 cm on 20 January 2022, and

the lowest value was 1.541 on 29 January 2022. A clear downward trend could be seen, except for two instances of PWV value increment on 25 January 2022 and 27 January 2022.

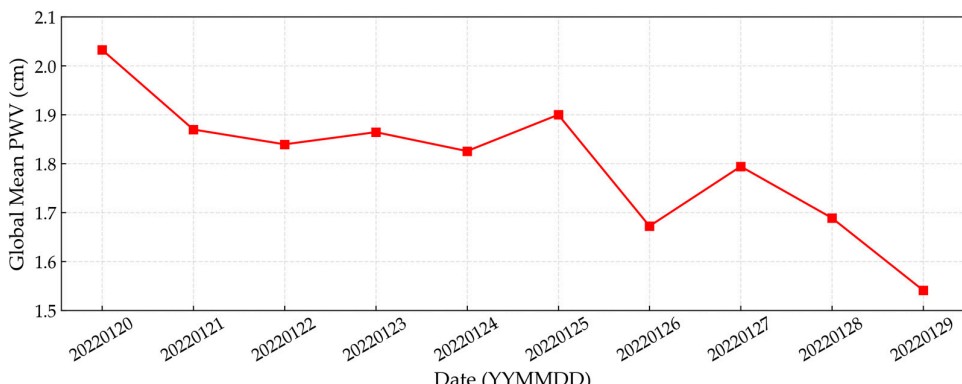

**Figure 5.** Global mean PWV variation from 20 January 2022 to 29 January 2022.

*3.2. Accuracy Assessment*

In this study, AERONET PWV data were used to evaluate the accuracy of DPC PWV data. The matching of AERONET PWV data and remote sensing PWV data was achieved according to the spatial and temporal matching method: The temporal mean of AERONET PWV data within half an hour of satellite transit was used to match the spatial mean of remote sensing PWV data in the range of $10 \times 10$ km$^2$ centered on the ground station [38]. We used three metrics to evaluate the accuracy of remote sensing PWV data, including mean absolute error (MAE), mean bias (MB), relative error (RE), and the coefficient of determination ($R^2$). The MAE was used to characterize the absolute error of remote sensing PWV data, the RE was used to characterize the relative error of remote sensing PWV data, and the MB was used to characterize whether the remote sensing PWV data were overestimated or underestimated. The calculation formulas of MAE, MB, and RE are as follows:

$$MAE = \sum_{i=1}^{N} \left| P_i - P_i' \right| / N \tag{5}$$

$$MB = \sum_{i=1}^{N} \left( P_i - P_i' \right) / N \tag{6}$$

$$RE = \sum_{i=1}^{N} \left| P_i - P_i' \right| / \sum_{i=1}^{N} P_i' \tag{7}$$

where $i$ represents a serial number, $N$ represents the number of valid matching results between remote sensing PWV data and AERONET PWV data, $P_i$ represents remote sensing PWV data, and $P_i'$ is the date corresponding to AERONET PWV data related to $P_i$. $R^2$ indicates the correlation between satellite PWV and AERONET PWV. To more comprehensively evaluate the accuracy of the DPC PWV data developed in this study, this study not only used ground-based data to verify the DPC PWV data but also validated the accuracy of MODIS PWV data released by NASA for comparison. The verification results of DPC PWV data and MODIS PWV data are shown in Figure 6. The MAE and RE of DPC PWV data were 0.32 cm and 0.30, respectively, and the MAE and RE of MODIS PWV data released by NASA were 0.23 cm and 0.26, respectively. The DPC and MODIS PWV results presented an overestimation situation with MB values of 0.29 and 0.21, respectively. Both DPC PWV and MODIS PWV had a high correlation with AERONET PWV, with $R^2$ values of 0.93 and 0.96, respectively. In general, DPC PWV data and MODIS PWV data had similar accuracy, with relatively close absolute error and relative error values. This meant that the overall accuracy of DPC PWV data was close to that of MODIS PWV data released by NASA. Since MODIS PWV data are high-precision PWV data [23], it can be considered that the DPC PWV data developed in this study have high accuracy.

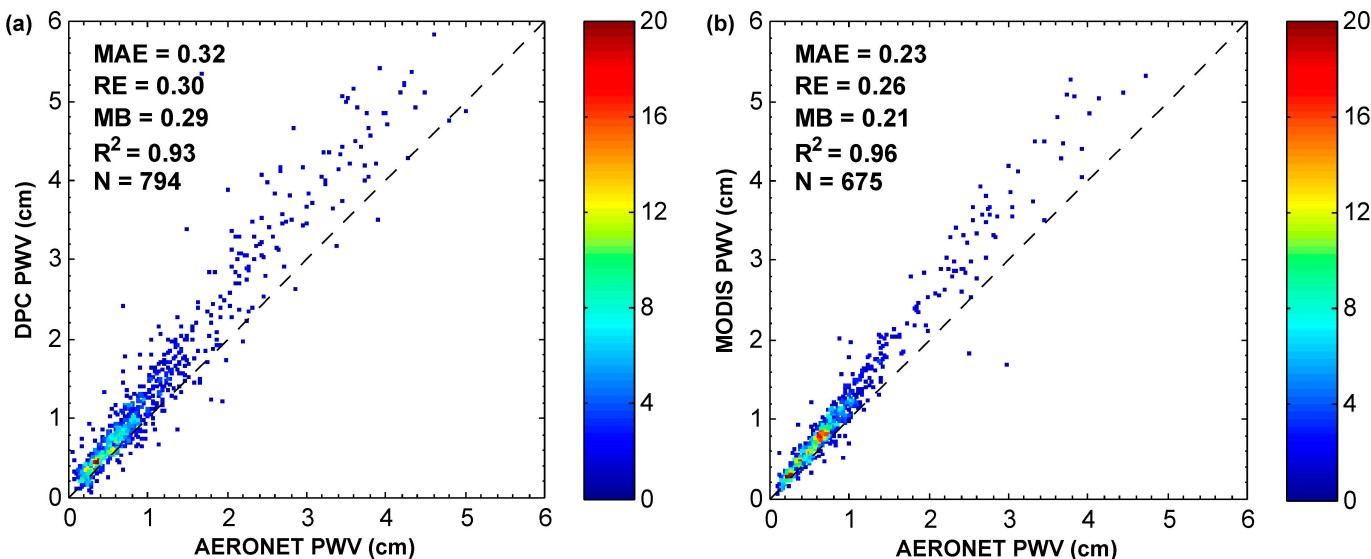

**Figure 6.** Validation results of satellite PWV data based on AERONET PWV data: (**a**) validation results of DPC PWV data; (**b**) validation results of MODIS PWV data. The color of each point denotes the number density at its location. The dotted line is the 1:1 line.

## 4. Discussion

Although the PWV data developed based on DPC data in this study have a low absolute error and relative error, the DPC PWV data still have some room for improvement. We analyzed the validation results and found that the DPC PWV data had a distinct overestimation trend (MB = 0.29 cm). To further explain this bias, we analyzed the DPC TOA reflectance in the 865 nm and 910 nm channels. We removed the absorption effect of water vapor on the 910 nm channel to obtain the corrected TOA reflectance, using the 6SV model with AERONET PWV data as input. Then, we compared the TOA reflectance in the 865 nm band with the corrected TOA reflectance in the 910 nm band. Figure 7 illustrates this comparison. Previous studies have shown that the TOA reflectance at 865 nm and the 910 nm TOA reflectance after water vapor absorption correction should be equal [37]. However, a distinct and systematic deviation is shown in Figure 7, which is inconsistent with the previous research conclusions. Since the DPC data used in this study were obtained during the on-orbit test period, the above inconsistency should be mainly because the calibration coefficients of the DPC 865 nm and 910 nm channels were not updated in time. A more detailed analysis of Figure 7 shows that the TOA reflectance in the DPC 865 nm band was systematically greater than the TOA reflectance in the DPC 910 nm band after water vapor absorption correction, which caused the WVAT in the DPC 910 nm band to be underestimated, resulting in an overestimation of PWV retrieval. Therefore, to further improve the accuracy of DPC PWV data in the future, it is necessary to update the calibration parameters of DPC 865 nm and 910 nm channels in time.

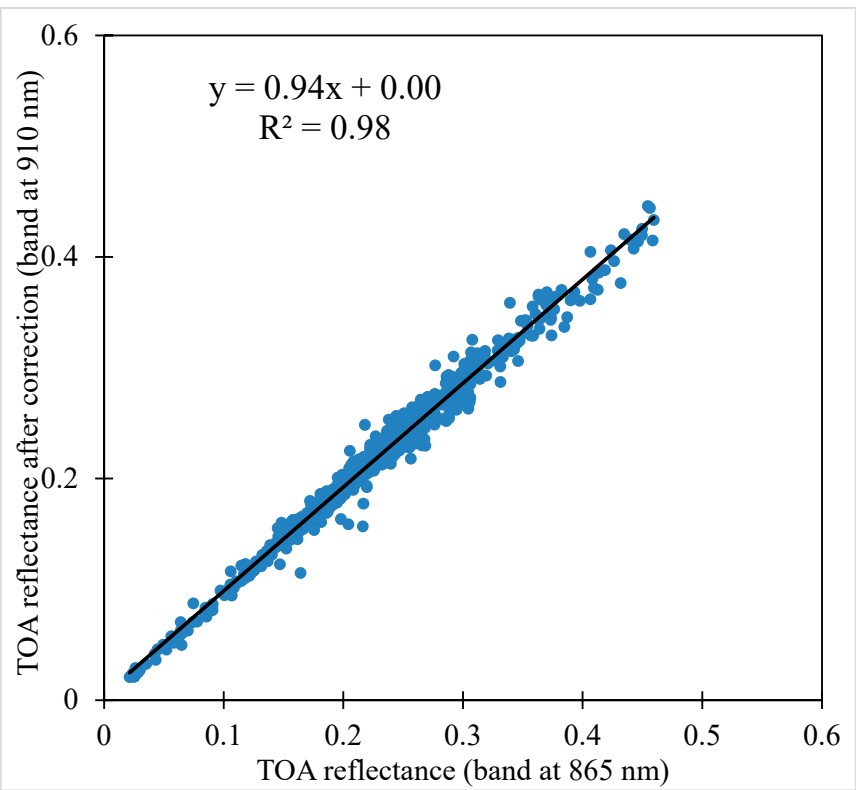

**Figure 7.** Comparison between DPC TOA reflectance in the 865 nm band and DPC TOA reflectance in the 910 nm band after water vapor absorption correction. The correction is implemented using 6SV with AERONET PWV as input.

## 5. Conclusions

The DPC is one of the main payloads on board the newly launched GF5-02 satellite. At present, the DPC is in the stage of on-orbit testing, and no publicly released DPC PWV data are available. In this study, we developed a DPC-suited water vapor retrieval algorithm based on the spectral characteristics of DPC data. The DPC water vapor retrieval algorithm consists of three parts: (1) the construction of the PWV retrieval lookup table, (2) the calculation of the WVAT in the DPC 910 nm band, and (3) DPC water vapor retrieval. The DPC PWV data are spatially continuous and can reflect the global water vapor content distribution. The validation results based on ground-based data show that the DPC PWV data developed in this study have metric values close to those of the widely used and accurate MODIS PWV data published by NASA. Although the DPC PWV data developed in this study have low absolute error and relative error, we still found a systematic overestimation trend in DPC PWV data. We found it was mainly caused by the calibration error. The DPC needs an updated calibration coefficient to further improve the accuracy of PWV products in the future.

**Author Contributions:** This work was carried out in collaboration with all the authors. Conceptualization, C.W., Z.S., Y.X., Z.L. D.L. and X.C.; Methodology, C.W., Z.S., Y.X. Z.L. and D.L.; Data acquisition, Z.S. and Y.X.; Supervision, Z.S. and Y.X.; Visualization, C.W., Z.S., Y.X. and D.L.; Validation; C.W. and Y.X.; Original draft preparation, C.W. and Y.X.; Writing—review and editing, Z.S., Y.X., Z.L., D.W. and X.C. All authors have read and agreed to the published version of the manuscript.

**Funding:** This work was supported by National Key R&D Program of China (grant No. 2020YFE0200700), National Natural Science Foundation of China (grant No. 42175147), and National Outstanding Youth Foundation of China (grant No. 41925019).

**Data Availability Statement:** The DPC data can be obtained from China Centre for Resources Satellite Data and Application (available online: http://www.cresda.com (accessed on 20 February 2022)).

The AERONET data can be downloaded from the official website of AERONET (https://aeronet.gsfc.nasa.gov/, last accessed on 5 March 2022). The MODIS data can be downloaded from the official website of NASA (https://ladsweb.modaps.eosdis.nasa.gov/, last accessed on 25 February 2022).

**Acknowledgments:** The authors thank Jun Lin, Zhongzheng Hu, and Lanlan Fan from China Centre for Resources Satellite Data and Application for providing DPC L1 data during the on-orbit testing of the Gaofen5-02 satellites. We would also like to thank the PI(s), Co-I(s), and their staff for their efforts in establishing and maintaining the AERONET sites for providing valuable validation datasets. In addition, we thank NASA for providing us with MODIS water vapor data.

**Conflicts of Interest:** The authors declare no conflict of interest.

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
