# Peer review of "Precipitable Water Vapor Retrieval Based on DPC Onboard GaoFen-5 (02) Satellite"

_remotesensing, doi:10.3390/rs15010094_

Round 1

Reviewer 1 Report

The manuscript by Wang et al., presents precipitable water vapor retrievals from GF-5(02) DPC. The employed algorithm is based on physical-based retrieval framework and the results seem valid with a validation with MODIS and AERONET. The paper is overall well-written and only a few issues should be addressed in the revised manuscript:

1) Please describe and address the main improvements of your algorithm compared to other existing algorithms.

2) The English language needs an improvement, a number of grammatical errors were found, like Line 27 "has high accuracy".

3) Subsections for Section 2.2 should be organized in a better way, Sect. 2.2.4 should be included in Sect. 2.2.1. as both subsections describe the retrieval procedure.

4) Figure 3: The title of Figure 3 indicates the ending date different from that in the text, please clarify it. What about blank areas in Fig. 3? No data? Please explain it in the text.

5) Line 282-284: Please provide an evidence (any reference) for your statement that MODIS has a high precision.

6) Figure 6:  Please indicate the corresponding R2 value. What does the color bar represent?

Author Response

Dear Reviewer,

        We sincerely thank you for the valuable feedback that we have used to improve the quality of our manuscript “remotesensing-2093604”. The suggestions are quite helpful, and we have incorporated them in the revised version of the manuscript. We hope that you will be satisfied with our responses to the ‘comments’ and the revisions for the original manuscript.

Thanks so much!

Chao Wang

Reviewer 2 Report

line 54    … DPC to retrieve aerosols …

line 65  why plural (bands)? I thought you only use the 910 nm band for PWV retrieval . Please clarify.

line 74  … which retrieves water vapor according to ….

line 99  do you mean 865 nm?

line 112    … spectral channels with a spectral range from visible to …

line 118    … to retrieve water vapor …

Figure 1  : please provide more informations: what is SRF/TTA ?
                  what does the T in WVAT mean?

line 133  : please explain why you introduce MODIS PWV. Do you need these data for intercomparison with DPC PWV?  Maybe it is good to provide at the end of the introduction an overview about the strategy of your article.

line 162   … PWV has an uncertainty of about 10% …

line 183    remove one time  „ to calculate the“ since it is doubled

Figure 3  line 233,  … for January 20, 2022  to January 29, 2022.

line 235  most regions are dry …

line 240   … the PWV values are high.

line 245    The 10 days zonally averaged PWV values observed by DPC

line 248  … of the southern summer hemisphere

Figure 5  please add the y-label including unit

line 283  does it mean that Aeronet is underestimating high PWV values in Figure 5 ?

 Figure 6   what is the label of the color table?

4. Discussion line 291   According to figure 6 , MODIS PWV is also overestimated at high values.
On the other hand you told that MODIS is accurate. Please clarify.  

Author Response

(The authors gave the same response as above.)
